# Methods of Observing the Signs of Approaching Calving in Cows—A Review

**DOI:** 10.3390/ani15071018

**Published:** 2025-04-01

**Authors:** Daria Wojewodzic, Marcin Gołębiewski, Grzegorz Grodkowski

**Affiliations:** Institute of Animal Science, Warsaw Univeristy of Life Science, Ciszewskiego 8, 02-786 Warsaw, Poland; daria_wojewodzic@sggw.edu.pl (D.W.); marcin_golebiewski@sggw.edu.pl (M.G.)

**Keywords:** calving behavior, artificial intelligence, cattle, prediction, imagine analysis

## Abstract

Predicting the exact date of calving is difficult due to varying gestation lengths and environmental factors. Missing the signs of calving can lead to complications, increased calf mortality, and financial losses. Monitoring methods can be invasive, for example, temperature loggers and blood tests, or non-invasive, such as sensors and video systems. Non-invasive methods improve cow comfort, while AI and machine learning have improved prediction accuracy, helping farmers manage calving more effectively.

## 1. Introduction

Calving is both the essence and the culmination of the milk production system. It is the moment when cows initiate lactation and ensure the future replenishment of the herd [1]. Unfortunately, predicting the exact time of birth is difficult. This is due to the variable length of cows’ gestation, the impact of the environment, and the uncertainty of the timing of calving [2]. The time of calving should be 268 to 296 days after successful insemination [3]. However, monitoring individual animals in large herds of cows during the perinatal period is particularly difficult. Failure to recognize the signs of impending calving and correct any calving-related problems can have disastrous consequences for animal welfare and productivity [2]. Dystocia is a difficult, prolonged labor, that may require veterinary intervention. It can be caused by a narrow birth canal, weak contractions, or the calf being too big [4,5]. Moreover, dystocia is a major problem in dairy cattle, with incidences ranging from 10.7% to 51.2% in the USA and 2.0% to 22.0% in Europe [6]. Also, perinatal mortality (stillbirths) is still very high in confined systems, ranging from 5.0% to 9.6% [7]. Farmers can reduce the risk of dystocia through management and supervision during calving [8]. Cattle often exhibit prolonged periods of inactivity before giving birth, especially in the last 6 h before calving, while additionally abstaining from feeding for over 2 h, which is often connected with difficulties in the birthing process [9]. Dystocia is recognized as a painful event. As a result of dystocia, a cow will often arch her back because she is in pain and stress. Pain evaluation and relief still need more attention from veterinarians and farmers [10,11]. The occurrence of dystocia is associated with the increased mortality of newborn calves and has negative effects on both dams and offspring [12,13]. The main reasons for monitoring the approach of calving in cows are prolonged delivery times, stillbirths, complications after calving related to impaired reproductive capacity, and prolonged intervals between calving and the next fertilization [14]. It is also important for calves to drink colostrum within 6 h after birth. The placenta of ruminants is synepitheliochorial, which prevents the transfer of immunoglobulins from the dam to the calf during pregnancy [15]. Newborn calves receive passive immunity after birth by absorbing immune-related cells and proteins from colostrum [16]. Unfortunately, dystocia also results in financial loss due to the impaired absorption of immune globulins from colostrum and increased calf mortality [12,17].

The onset of calving is very difficult to detect; therefore, it is crucial to understand behavioral changes. It is important to note that the birth of a healthy calf is one of the economic successes of cattle production [18]. Calving in cows occurs in three stages [18,19,20]. During the first stage of birth, the cervix relaxes and dilates [18,20]. In the second stage of calving, contractions of the uterus continue, then the fetus enters the birth canal, followed by the expulsion of the fetus [18,20]. The last stage of calving is the expulsion of the fetal membranes [20]. It is noteworthy that physiological change before calving is a decrease in body temperature, which results from increased placental blood flow in the immediate pre-calving [21]. Miwa et al. [22] indicate that the cow’s body temperature begins to decline approximately 48 h before calving. Specifically, the temperature of the abdominal region and the vagina decreases between 32 and 36 h before parturition. Additionally, within the final 12 h preceding calving, the body temperature of the cow decreases by approximately 0.6 °C. Several endocrinological changes also occur before calving. Estrogen levels gradually rise between 20 and 10 days before giving birth [23]. The peak level of these parameters is reached between four and one day before delivery (2.9 ng/mL for estrone, 1.4 ng/mL for estradiol-17β). Conversely, progesterone levels drop sharply in the last 36–38 h before calving. A rapid increase in progesterone occurs 24 h before calving and reaches a level of about 5.7 ng/mL [23]. The last 24 h before calving are marked by an increase in lying frequency, walking frequency, and tail-raising frequency in dairy cows [24], while rumination time decreases between 24 and 12 h before calving [25].

Calving is associated with various physiological changes, for example, increased production of corticotropin and the conversion of progesterone to estrogen. Estrogen stimulates the production of prostaglandin, which induces cervical relaxation and myometrial contractions [20]. Additionally, oxytocin is released and blood glucose levels increase by 42.2%, as demonstrated by Zamaziy et al. [26]. A significant indicator of impending calving is the decrease in vaginal temperature by approximately 0.2–0.3 °C [27]. Measuring a temperature decrease in the vagina of about 0.3 °C or more can help predict calving occurring within 24 h. Furthermore, there is a relaxation of the uterine ligament, swelling of the uterus, and enlargement of the mammary gland.

In addition to physiological changes, calving is also associated with behavioral changes. Cows build a nesting area and often look at the back of their abdomens [18,28]. Before calving, cows spend additional hours standing, and the frequency of transitioning from lying to standing increases by 80.0% [29]. Pre-calving behavioral changes include increased restlessness, and isolation-seeking, and are associated with frequent postural changes [30,31]. Furthermore, lying behavior differs between multiparous and primiparous cows. Primiparous cows lie down more frequently but for shorter periods of time than multiparous cows [32]. Moreover, there are decreases in food intake and in ruminating. Food intake and ruminating gradually decrease about two weeks before calving. This is associated with a decrease in dry matter intake [33]. Primiparous cows have lower dry matter intake, spend more time feeding, and visit feeders more frequently [32]. About 6 h before calving, tail lifting increases significantly [34]. Lying with extended legs is a common posture for cows shortly before calving; this typically occurs within the final hour. However, if a cow remains in this position for more than two hours, it may indicate potential calving complications [9].

In large herds of cows, the monitoring of a single individual is difficult [2]. The two main problems associated with the calving period are dystocia and perinatal mortality [21]. The aim of this article is to gather information about both invasive and non-invasive methods for observing the signs of impending calving. The beginning of calving is very difficult to observe because there are a lot of behavioral and physiological changes. It is only during the second stage of birth that we can observe the amniotic sac. Failure to recognize the signs of impending calving is connected with economic consequences such as reduced productivity and herd culling [35].

At the outset, collecting data from cattle farms was based on people observing and recording the most important information in notebooks. With the development of digitalization, all the data were recorded on computers and high-capacity disks. Human observation began to be replaced by installed video cameras. The first sensors used to identify individual animals appeared in 1970 [36]. Expert systems in the form of computer software applications used for the inference and analysis of behavior, physiological parameters, and recognition of individuals appeared in the 1980s [37,38]. The introduction of radio frequency identification (RFID) tags for livestock in the early 2000s was a significant step forward [39,40]. This technology has established a rudimentary system for identifying and tracking individual animals and has allowed for the development of artificial intelligence. Currently, AI allows for the study and interpretation of advanced parameters concerning the risk of ketosis, genomic research, milk production, fertility, and nutrition, as well as predicting the approach of calving [41].

## 2. The Use of Artificial Intelligence (AI) in Cattle Breeding

Methods used for observing the approach of calving can be divided into invasive and non-invasive. Invasive methods include temperature loggers in the uterus [27], the intrauterine placement of GSM (global system for mobile communications) devices [14], testing electrolyte concentrations from udder secretions [2], and blood sampling to test progesterone levels [42]. With the development of technology, invasive methods have been supplanted by non-invasive methods. This provides greater comfort and welfare for the cattle. Non-invasive methods include transponders mounted on the neck, the use of real-time monitoring to evaluate locomotion, recorders placed on the ears, and sensors mounted on the limbs or tail [43]

Observing the signs of impending calving, especially using non-invasive methods, is particularly important because, as found in a study by Lombard et al. [12], 51.2% of calves born to primiparous dams and 29.4% of calves born to multiparous dams require assistance during calving.

## 3. Invasive Methods for Observing the Signs of Impending Calving

A method commonly used to identify illness in dairy cows is measuring body temperature with a rectal thermometer, but vaginal measurements are becoming common in research [44]. Furthermore, physiological status (e.g., estrus, lactation, pregnancy, and calving) influences the diurnal temperature patterns of dairy cows [45,46]. In their research, Burfeind et al. [27] used temperature loggers (Minilog 8, Vemco, Ltd., Halifax, NS, Canada; Table 1) that were inserted into the vagina of cows before calving. The vaginal temperatures were 0.2 °C to 0.3 °C and 0.6 °C to 0.7 °C lower on the day of calving compared with 24 h and 48 h before calving. The team of scientists also studied changes in rectal temperatures. The examined readings were 0.3 °C to 0.5 °C and 0.4 °C to 0.6 °C lower on the day of calving compared with 24 h and 48 h before calving. A decrease in vaginal temperature of >0.3 °C and, similarly, a decrease in rectal temperature of >0.7 °C may predict calving within 24 h. The effectiveness of measuring the vaginal temperature is shown in Table 1. Unfortunately, temperature loggers are affected by the environment and changes in temperatures in cowsheds. On the other hand, dairy cows exhibit a distinctive decrease in body temperature approximately 48 h before calving. Detecting this decrease can provide additional information about the onset of calving.

Measuring temperature in the vagina is also used in beef cattle to detect calving time. Aoki et al. [47] used a data-logging apparatus (Data-logger L820, Unipulse Inc., Nishinasuno, Japan) with a thermocouple sensor in beef cows with twin fetuses. The devices enable effective monitoring of twin-bearing pregnant cows, enhancing the calving process while facilitating a deeper understanding of the physiological changes occurring in beef cows during the late stages of twin gestation.

Tracking changes in body temperature provides many opportunities to detect changes prior to calving. Costa et al. [48] observed changes in temperature that occurred in the reticulorumen. Each cow received an orally administered, temperature-sensing reticulorumen bolus (Phase IV Engineering, Boulder, CO, USA; Table 1). The results indicated that a drop in the reticulorumen temperature of about 0.2 °C was associated with the onset of calving [44]. The reticulorumen bolus may be a useful tool in creating a calving prediction algorithm. Also, Cooper-Prado et al. [49] used ruminal boluses (SmartStock, LLC, Pawnee, OK, USA) to evaluate changes in ruminal temperature, which decreased 1–2 days before parturition (38.88–38.55 °C). Metabolic adaptation and endocrine and behavioral changes during the periparturient period may cause a decrease in ruminal temperature [50]. The measurement of ruminal temperature may result in management systems that can increase the reproductive performance of beef cows.

Another method of detecting impending calving is to use an intra-vaginal GSM device, with the sensor being inserted into the vagina before the expected calving. The intra-vaginal device is pushed out at the second stage of calving and activates a radio transmitter [14]. This system is a tool that supports farm workers in reducing the negative factors of dystocia and helps prevent stillbirths. This sensor reduces the negative impacts of dystocia by improving cow welfare and increasing calf vitality. The use of remote calving monitoring devices accelerates the reaction time of qualified personnel. Intra-vaginal GSM devices enhance calving supervision [26].

Another proven and fast method is the on-farm blood progesterone enzyme immunoassay (EIA) test [42], which is commercially available (CITE PROBE Semi-QuantTM Progesterone, Idexx Corp., Portland, ME, USA; Table 1). Matsas et al. [42] found that more than 95.0% of cows calved within 24 h when progesterone levels were <1.3 ng/mL. The sensitivity, specificity, and predictive value of the EIA in determining parturition within 24 h were 86.7%, 90.8%, and 75.0%, respectively. Predicting the onset of parturition by determining progesterone levels through enzyme immunoassay is also used in sows [50] and goats [51]. Measuring progesterone in dairy cows is a basic and applied observation method used in reproduction physiology [52]. A commercially available semi-quantitative EIA kit (Hormonost Easy Rind; Biolab, Oberschleißheim, Germany) was used to analyze blood plasma [53]. EIA kits are particularly used in difficult conditions by veterinarians and field workers [53]. The test was 90% accurate in determining the day of parturition [54]. Based on the EIA result it was predicted that the cervical dilation was attributable to the initiation of the first stage of parturition and a decision was made to monitor cervical dilation [54]. The most important information for farm management is the ability to predict parturition within 12 h after examination [53]. Therefore, the EIA kit is an excellent tool to optimize calving management in cows because the ability to predict parturition within 12 h in cows was 99.3% [53]. Testing for progesterone levels in the blood is an easy and fast method, especially when detecting the onset of labor in dairy cows in pastures where there is no access to specialized technology. Rapid progesterone tests provide the ability to detect the first phase of labor, and combined with monitoring of cervical dilation, provide the opportunity to respond properly in the developing phase of labor. The use of the progesterone kit is a valuable aid in clinical management [54].

In addition to testing blood samples, it is also possible to test secretion samples from the mammary gland. Research was carried out on concentrations of calcium, inorganic phosphorus, chloride, magnesium, sodium, and potassium using photometric methods [2]. In addition, in research by Bleul et al. [2], the concentration of calcium, magnesium, and inorganic phosphorus was determined using rapid test kits. The highest correlation between impending calving and electrolyte level was for organic phosphorus (r = 0.74; 2). The inorganic phosphorus concentration was 11.8 mmol/L to 26.5 mmol/L in cows that calved within 24 h of sample collection [2]. The phosphate test is straightforward and easily carried out in the field by veterinarians or laypeople. It can help determine when to move cows to the calving pen and makes cow monitoring more efficient [2]. It is best that the test is performed in combination with temperature change observations in the cows.
animals-15-01018-t001_Table 1Table 1Comparison of the effectiveness of impending calving detection using invasive methods.Name of DeviceMeasured ParameterBeef or Dairy CowsSensitivitySpecificityLiterature SourceMinilog 8Vaginal temperatureDairy cows62–71%81–87%[27]Data-logging apparatus (Data-logger L820, Unipulse Inc., Nishinasuno, Japan) with a thermocouple sensorVaginal temperatureBeef cows--[46]Intra-vaginal deviceSecond stage of calvingDairy cows--[14]CITE PROBE Semi-Quant^TM^ ProgesteroneBlood progesterone levelsDairy cows86.7%90.8%[42]Inorganic Phosphorus, Ref. A11A00098; ABX Diagnostics, Dietikon, SwitzerlandPrepartum mammary secretionDairy cows--[2]Reticulo-rumen bolus (Phase IV Engineering, Boulder, CO, USA)Reticulo-rumen temperatureDairy cows58–73%63–77%[47]Ruminal boluses (SmartStock, LLC, Pawnee, OK, USA)Ruminal temperatureBeef cows--[49]

## 4. Non-Invasive Methods for Observing the Signs of Impending Calving

In dairy farming, it is very important to use precision dairy monitoring technologies to provide alternative observation and assessment of calving behavior. These represent alternative approaches for predicting calving time. AI can be utilized to monitor activity, chewing, lying time, and walking in order to characterize prenatal behavior. The use of video cameras and remote monitoring of cows allows the negative human factor that occurs in the natural environment of cows to be minimized. Close observation of cattle using modern technology during the last gestation period is essential for detecting the onset of calving and, therefore, reducing neonatal losses [55]. Unfortunately, up to one-third of calves born on dairy farms are born after dystocia and have increased risks of disease and mortality [56]. That is why it is so crucial to use automated measurements for lying time, number of lying periods, and ruminating; this will help prevent the negative effects of calving.

In the final 6 h period before calving, cows spend less time lying down, and more time standing, and demonstrate physical activity [57]. Moreover, there is a decrease in physical activity at about 2–4 h before birth. Cows adopt a lateral lying position with their heads loosely resting on the ground [57,58]. The locomotion and posture behavior of pregnant cows can be used as a form of observation for signs of impending calving. Cangar et al. [59], in their study, investigated an automatic real-time monitoring technique. Real-time monitoring is a non-invasive method of observing cows. In the study, the behavior of cows was observed and classified: standing and walking time, back position, and walking distance. Studies have shown that 1 h before calving, cows prefer to stand in the corner of their box, whereas in the previous hours before giving birth, cows are active and move in circles [59]. Such behaviors can be analyzed through the application of artificial intelligence, which enables the detection of a cow’s posture. Giaretta et al. [60] used a real-time video camera and artificial intelligence to detect cow behaviors and postures, for instance, tail raising, walking around in the birthing box, standing, lying, and eating [60]. The use of a camera provides farmers with automatic tools to monitor cows in their boxes, annotate movements, and change behavior [61].

Pronounced laterality may indicate that cows are uncomfortable. Cows that are in late pregnancy show marked laterality while lying, perhaps because of discomfort associated with the calf [62]. On this account, lying behavior is a good indicator for observing the signs of approaching labor. Ledgerwood et al. [63] evaluated the HOBO Data Logger. These loggers were attached to the left legs of the cows and measured lying behavior. Automated measurements of lying time, the number of lying periods, the structure of these periods, and the laterality of lying behavior will all reduce labor requirements for this type of research and improve our biological understanding of cow comfort [63]. Mattachini et al. [64] validated the HOBO Data Logger (Table 2). The sensitivity and specificity for combining the dataset and behavior were high. The HOBO device accurately measured lying and standing behavior in cows [64].

Loggers on the legs or collars attached around the neck of cows can be used for observing signs of impending calving in cows. SCR HR and LD tags (SCR Engineers, Ltd., Netanya, Israel, Table 2) were used in a study conducted by Clark et al. [65] to determine rumination duration profiles and levels of activity; these data were then used to predict the day of calving. In the Clark et al. [65] study, rumination duration for cows decreased by 33% over the two days prepartum. In contrast, activity levels were maintained prepartum but increased in the days postpartum [63]. In a later study, Borchers et al. [19] used automated activity monitors to measure lying and rumination activity in cows, in order to characterize postpartum behavior and predict calving in dairy cattle. They also used HR Tags and, additionally, the IceQube sensor (IceRobotics Ltd., South Queensferry, UK; Table 2), which automatically collected number of steps, lying time, standing time, number of times transitioned from standing to lying, and the total motion of the cows. The optimal lying duration for cows is between 12 and 14 h per day [66,67]. Lying time decreased by 8 h before calving in this study. On the other hand, neck activity and number of steps were no different up until the day before calving [19]. Daily rumination time decreased through the prepartum period and was at its lowest level on the day before calving.

Another way to observe the signs of impending calving, apart from loggers attached to the legs, is a noseband sensor. In a study by Mahmoud et al. [68], cows were fitted with the RumiWatch noseband sensor (ITIN + HOCH GmbH, Fütterungstechnik CH-4410, Liestal, Switzerland; Table 2) along with a 3D accelerometer attached to the hind limbs. The devices measured ruminating time, number of ruminating chews, number of ruminating boluses, locomotion behavior, and number of lying periods. In primiparous and multiparous cows, the number of lying periods increased in the last 3 h prior to calving [69]. The relative value of the number of rumination boluses per three-hour period decreased significantly in the last three hours before parturition in multiparous cows. Furthermore, the relative value of the number of ruminating chews per three-hour period decreased significantly in the last three hours before calving in primiparous cows [69]. The RumiWatchSystem noseband sensor and 3D accelerometer are tools that can be used to predict the onset of calving within the next 3 h with a high level of accuracy [57] (Table 2).

An important factor in indicating the approach of calving is rumination. Bar and Solomon [70] reported a decrease in daily rumination time of around 255 min on the day of calving. Similar results were reported by Soriani et al. [71], who found a reduction in daily rumination time in the days leading up to calving. Rumination time can be an indicator for detecting calving events or health disorders. Rutten et al. [8] attached Agis SensOor sensors (Agis Automatisering B.V., Harmelen, The Netherlands) to cows’ ears. The sensor tag measured rumination, feeding, and temperature (Table 2). The Agis SensOor was validated by Bikker et al. [72]. In this study, an increase in activity and a decrease in rumination prior to calving was noted. Regarding temperature, a decrease of about 3 °C was observed during the hour in which calving started. However, a sensor attached to the ear is more sensitive to environmental influences [8]. In previous studies, the temperature measured in the vagina before calving had decreased by about 0.2–0.5 °C [27,73]. The decrease in rumination in the current study was about 15 min/h [8]. Changes observed in the behavior of cows before calving are crucial signs, so the farmer’s presence is important for observing these indicators and thus preventing dystocia.

One of the non-invasive methods for the detection of labor is measuring the temperature on the ventral side of the tail. The skin temperature is measured by a tail-attached sensor. Higaki et al. [74] created a calving prediction model based on data from tail sensors. This model could help predict dystocia calving [74]. Another solution for detecting parturition in cows using a device installed on the tail is the tail-mounted sensor. The tail-mounted sensor detects whether the tail is raised or lowered, which the sensor classifies as a contraction. It is envisaged that fewer false alarms and fewer missed births will result as a consequence of using this method. Yet another physiological parameter that may indicate signs of approaching labor is the glucose level; however, blood sampling in cows can cause stress. Wakatsuki et al. [75], in their study, used a wearable sensor. The tissue glucose concentration increased at about 2.8 h before calving [76]. Additionally, tissue glucose concentration was significantly higher in primiparous cows than in multiparous cows [75]. One of the latest studies to investigate tail movement sensors that are used to predict the time of approaching calving was presented by Umaña Sedó et al. [76]. The researchers studied and evaluated a sensor (Moocall; Bluebell, Dublin, Ireland; Table 2) to predict the time of calving in dairy cows. The time of calving was best predicted at 12 h before calving. From 12 h to 2 h before calving, the decrease in diagnostic accuracy of all the alarms was pronounced [76]. The decrease in the accuracy of the tail movement measurement by the sensor may have been due to the cow’s interest in the tool that was attached to their tail. This may have caused false results. Investigations on individual cows revealed an increased number of time slots showing a significant increase in the variability of activity, and an increased frequency of tail raising and rubbing the tail on objects in some cows after the calving sensor was attached—this should be investigated in more detail and on a larger scale [76].

It is worth noting that Rice et al. [77] conducted research on the prepartum lying behavior of Holstein dairy cows kept on pasture. IceTag data loggers were used to measure lying time, number of lying periods, duration of lying periods, and steps. In this study, cows had shorter mean daily lying time than cows in barns. This information may suggest that pasture had no negative effect on prepartum cow comfort in conjunction with reduced lying time [77]. There was also no increase in lying periods for cows housed on pasture. This indicates that cows housed on pasture were less restless and uncomfortable during the prepartum period than cows housed in barns. This suggests that even though cows were approaching calving they did not experience enough discomfort to affect lying-period duration [77,78].
animals-15-01018-t002_Table 2Table 2Comparison of the effectiveness of impending calving detection using non-invasive methods.TechnologyLocation on CowParameters MeasuredSensitivitySpecificityValidationPatentSCR HR LD Tags (SCR Engineers, Ltd., Netanya, Israel)Left side of neckRumination and level of activity70.0%70.0%[79]Inventor: Avshalom Bar-ShalomUS7350481B2The IceQube (IceRobotics Ltd., South Queensferry, UK)Left rear legNumber of steps, time spent lying, time spent standing, lying periods, total motion37.5–75.0%87.9–91.2%[19,64,80]
The RumiWatch system (ITIN + HOCH GmbH, Fütterungstechnik CH-4410, Liestal, Switzerland)Noseband sensor and 3D accelerometer on the hind limbsLocomotion behavior and number of lying periodsMultiparous: 88.9%Primiparous:85.0%Multiparous:93.3%Primiparous:74.0%[81]Swiss Patent CH 700 494 B1Agis SensOor sensors (Agis Automatisering B.V., Harmelen, The Netherlands)EarRumination, feeding, activity, and temperatureEvaluation in daily basis:9.1–36.4%Evaluation on hourly basis:21.2–51.5%Evaluation in daily basis:98.9–99.3%Evaluation on hourly basis:99.1–99.4%[72]
HOBO Data LoggerLeft rear leg (below hock)Lying behavior

[63,64]
Moocall©, Moocall Ltd., Dublin, IrelandTailTail movement

[60]
Tail-attached sensorTailSkin temperature84.3%
[82,83]
Tail mounted sensorTailRaising or lowering of the tail

[78]Inventor:Austin N., Vukajlovic M.WO2017211473A1FreeStyle Libre Sensor (Abbott Japan LLC, Chiba,Japan) and LibrePro Sensor (Abbott Japan LLC)Back side of the tailTissue glucose concentration71.9–77.8%74.9–83.4%[75]
Tail movement sensor device (Moocall; Bluebell, Dublin, Ireland)TailTail movement72.0%72.0%[76]
IceTag data loggers (IceRobotics, Ltd. Edinburgh, Scotland, UK)Hind legLying time, number of lying periods, duration of lying periods, steps

[77]


## 5. Review of the Possibilities of Using AI in Detecting Parturition in Cows

Calving difficulty in cows generates a higher probability of calf and cow death and also affects economic issues and veterinary costs. The above-mentioned solutions are mainly based on expert systems that help identify whether a cow is about to give birth. Technologies, especially invasive ones, have been replaced by AI systems. Sensors are required to be put inside or on the bodies of cows and this is a burden for the cow and causes stress [84]. Recent technological advances in the field of computer vision are based on the technique of deep learning and AI systems [85]. Data obtained from sensors, for example, accelerometers and thermometers, can be successfully used to train artificial intelligence. Computer vision combined with AI can be used for a number of animal monitoring tasks, such as recognizing the type of animal, detecting where animals are located in the image, and locating parts of their bodies [84]. Thanks to the development of AI, image analysis has become possible. Most behaviors can be recorded and recognized by AI and the use of RGB cameras. Nasirahmadi et al. [86] in their study, demonstrated that posture recognition can be achieved with high accuracy using RGB cameras and AI. For example, eating and drinking can be detected with a high level of accuracy of over 90.0%, thanks to the use of computer vision. These behaviors can be used to identify birth events and health problems [87]. Furthermore, standing, lying, eating, and drinking behaviors all scored greater than 84.0% and can also help with the monitoring of animal well-being [85]. Computer vision can be successfully applied to predict individual dairy cow behaviors. This approach could be used for the early detection of abnormal behavior in animals and birth detection [85]. Automated monitoring of cows during parturition is beneficial and assists the stock person and enhances animal welfare. The 3D camera can reduce the burden on both workers and the cows [84]. This also reduces the costs associated with taking on employees.

The development of artificial intelligence in agriculture is accelerating. More and more farms are using advanced technologies to monitor animals. The future of AI in birth detection includes the use of biosensors, which enable the measurement of physiological parameters such as body temperature, hormone levels, and motor activity. Temperature measurements are performed using vaginal sensors or boluses placed in the rumen. These tools show that cows’ temperatures drop by 0.2–0.5 °C within 24 h of giving birth [27]. Progesterone tests have shown greater than 85% efficacy in predicting calving within 24 h at concentrations below 1.3 ng/mL [42]. The analysis of physical activity is also an important measurement. Before calving, cows reduce chewing time and increase the number of position changes from lying to standing by more than 80% within 6 h before giving birth [29]. AI analysis in birth detection allows the use of 3D image recognition technology, which improves the accuracy of monitoring cow posture before birth. Artificial intelligence systems also enhance the precision of algorithms for Big Data analysis, enabling the effective detection of non-standard cases. Cameras that record the behavior of cows in barns facilitate the analysis of body posture, as well as head and limb movements. Before calving, cows are more likely to lie down, while tail lifting occurs more frequently about 6 h before calving [34]. Additionally, increased nervousness and isolation from the rest of the herd are characteristic signs of impending calving [30]. Moreover, machine learning (ML) algorithms are employed in data analysis. Machine learning-based models enable the automatic classification and interpretation of complex patterns of cow behavior before parturition. Data collected by sensors and vision systems are analyzed by AI algorithms, which can then predict the moment of delivery with an accuracy of over 90% [85].

Future lines of research on the detection of calf birth in dairy cows should focus on rumination behavior and cow activity during the day [88]. Artificial intelligence models should be adapted to commercial animal farming conditions. What is important is a practical analysis of the conditions that prevail on the farms, which will affect the processing of digital data: the dirt of the cameras, the temperature in the barn, the temperature of the drinking water, and the preparation of the feed [89]. Hogeveen et al. [89] defined three criteria that must be fulfilled for a detection model in commercial livestock production. First, high performance in terms of data sensitivity and specificity. Second, the time it takes for an artificial intelligence model to react and adapt to farm conditions. Third, a similarity between the research project and the everyday conditions on the farm. Furthermore, Dominiak and Kirstensen [90] found that over the past 20 years, sensor-based studies and detection models have not met the performance requirements that would generate a satisfactorily low level of false alarms. The authors specify that the detection of births should focus not on unattainable results of sensitivity and specificity, but on demonstrating the likelihood of the occurrence of the early signs of impending birth and on response time. The research reports [91] that the future of birth detection in dairy cows is linked to the use of increasingly new artificial intelligence algorithms. The authors evaluated RNN (repetitive nearest neighbor) algorithms [91]. Future research directions related to the observation of signals that indicate approaching calving will be based on LSTM (Long Short-Term Memory) algorithms [91].

Unfortunately, artificial intelligence has its limitations. The learning process is difficult and time-consuming and requires a large database as well as expert annotations. In the application of 3D image recognition technology, camera dirt and defects may occur. In addition, contrary to popular opinion, AI does not work with 100% accuracy and there is the risk of error. It is important that when optimizing artificial intelligence models, statistical parameters should not be limited, for example, sensitivity and specificity [88]. These parameters can reach high values and suggest that the performance of the model is appropriate.

## 6. Conclusions

In many areas of the world, the number of cows per dairy farm has increased and, as a consequence, the time farmers can spend on individual cows has been reduced [72]. The result has been to discover ways to detect an approaching calving as accurately as possible. It is worth noting that methods of observing an approaching birth should be as minimally invasive as possible. A good choice is to use precision breeding [92]. Precision farming is the real-time monitoring of animals through video analysis, data analysis, and analysis of physiological parameters. The successful application of AI can have a significant impact on the functioning of dairy farms [93]. Predicting calving is key to dairy cow management. Automated monitoring of behavioral and physiological changes prior to parturition can be used to develop machine learning solutions for calving prediction [94].

## Data Availability

All data generated or analyzed during the study are included in this published article. The datasets used and/or analyzed in the current study are available from the corresponding author upon reasonable request.

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
