# Peer review of "Methods of Observing the Signs of Approaching Calving in Cows—A Review"

_animals, 2025, doi:10.3390/ani15071018_

Round 1

Reviewer 1 Report

Comments and Suggestions for Authors

A review entitled Methods of Observing the Signs of Approaching Calving in 2 Cows presents approaches used in predicting the onset of parturition in cows. The authors describe and summarise both invasive and non-invasive methods allowing to know the changes in behaviour and physiological parameters just before calving. The devises that can be applied inside a birth canal of a cow or require blood collection are presented in relation to the time of parturition. The possibilities of measuring or observing changes in rumination, walking activity, lying , tail movements, and body temperature are presented with their advantages and limitations. The authors also describe how the present technologies (artificial intelligence etc.) could be used now and in the future to help to predict parturition or necessity of assistance during parturition.

Some minor mistakes and remarks should be corrected and implemented into the manuscript and they are listed below:

  • Line 38 – it would be useful to define dystocia, because can be understood differently (for example as any help during parturition, assistance of veterinarian or the other)
  • Line 60 – should be fetal membranes (placenta is an organ consisted of maternal and fetal parts. Expulsion refers only to fetal part.
  • Line 65 – level
  • Line 110 – please check citations of the publication numbered 18 throughout the manuscript. That publication presents different data than the authors cite
  • Line 256 – delete the sentence ‘The creators…”, just cite
  • Lines 305-321 – re-write this paragraph, there are repetitions, mistakes etc.

Author Response

Dear Reviewers,

On behalf of my co-author and myself, I would like to thank the reviewers for their valuable comments and suggestions. Below are detailed responses to comments on the manuscript. The corrections made are highlighted in the manuscript.

The manuscript was also proofread for linguistic correctness. The language certificate is sent with the manuscript

Sincerealy,

Daria Wojewodzic, Marcin Gołębiewski, Grzegorz Grodkowski

Comments 1: Line 38 – it would be useful to define dystocia, because can be understood differently (for example as any help during parturition, assistance of veterinarian or the other)

Response 1: The defeniction of dystocia has been added. Lines: 38-40

Comments 2: Line 60 – should be fetal membranes (placenta is an organ consisted of maternal and fetal parts. Expulsion refers only to fetal part.

Response 2: It was changed ‘’placenta” to ‘’fetal membranes”. Line: 66

Comments 3: Line 65 – level

Response 3: It has been corrected ‘’levels” to ‘’level”.

Comments 4: Line 110 – please check citations of the publication numbered 18 throughout the manuscript. That publication presents different data than the authors cite

Response 4: Thank you for pointing this! I checked citation through the manuscript.

Comments 5: Line 256 – delete the sentence ‘The creators…”, just cite

Response 5: The sentence has been removed.

Comments 6: Lines 305-321 – re-write this paragraph, there are repetitions, mistakes etc.

Response 6: The paragraph was rewrited.

Reviewer 2 Report

Comments and Suggestions for Authors

The review article investigates the role of technologies in detection of the onset of parturition. This is a relevant and timely review, with the use of technologies becoming more commonplace on farm and with calving being a critical event. However, the review lacks clear organisation and tends to repetitive. It would benefit from a more structured presentation. The review would be more informative if it included more data to support the discussion. Rather than simply stating changes in the levels of the parameters discussed, it would be useful to include what constitutes the normal range. The reader can then calculate the expected levels during calving for themselves. 

Some examples of the issues that i perceive are given below:

Line 42: Cattle often exhibit periods of inactivity - when

Line 48: Many dairy farmers would like to know when a cow will calf for better post-natal management of the calf. 

Line 50: Fertilisation- do you mean the next conception, i.e. calving to conception interval?

Line 51: It is important for calves to drink colostrum by 6h ....

Line 60: several changes in endocrine parameters are listed, with limited discussoin of physiology

Line 65: why is it important that body temperature decreases?

Line 77: Changes - be specific, increases or decreases.

Line 79: dry matter, rather than dry mass.

Line 80-82: these are physiological changes, not behavioural changes.

Line 82: lying, not sitting.

Line 87-88: repeating what has been said earlier in the review.

Line 95: choice of words -  At the beginning???

Line 119- 121: repeating what has been said earlier.

Line 138_140: Discussion of body temp would be better earlier, before discussion of changes in temp.

Line 151-152: How does this sensor reduce the negative impacts of dystocia?

Line 160- Measuring progesterone requires a blood samples - not possible on all farms.

Line 162-169: Is it a good idea to take strip samples before calving?

Line 192: could this change in behaviour be automated.

Line 195 - 220: What are normal lying times in cattle?

Line 262-263: changes in glucose concentration post calving - too late to be a predictor of calving?

Line 212: Borchers et al is reference 14.

Comments on the Quality of English Language

The English is generally fine

Author Response

Dear Reviewers,

On behalf of my co-author and myself, I would like to thank the reviewers for their valuable comments and suggestions. Below are detailed responses to comments on the manuscript. The corrections made are highlighted in the manuscript.

The manuscript was also proofread for linguistic correctness. The language certificate is sent with the manuscript

Sincerealy,

Daria Wojewodzic, Marcin Gołębiewski, Grzegorz Grodkowski

Comments 1: Line 42: Cattle often exhibit periods of inactivity – when

Response 1: It was mentioned that the period of decline in activity in cows occurs 6 hours before birth. Lines: 45-46

Comments 2: Line 48: Many dairy farmers would like to know when a cow will calf for better post-natal management of the calf.

Comments 3: Line 50: Fertilisation- do you mean the next conception, i.e. calving to conception interval?

Response 3: Yes, I mean the next conception i. e. calving to conception interval.

Comments 4: Line 51: It is important for calves to drink colostrum by 6h ....

Response 4: It was explained why colostrum is important for calves. Lines: 55-58

Comments 5: Line 60: several changes in endocrine parameters are listed, with limited discussoin of physiology

Response 5: Physiological changes, especially in hormone levels, were mentioned. Also added a discussion on body temperature. Lines: 67-82.

Comments 6: Line 65: why is it important that body temperature decreases?

Response 6:

Comments 7: Line 77: Changes - be specific, increases or decreases.

Response 7: Has been listed – decreases. Line: 85

Comments 8: Line 79: dry matter, rather than dry mass.

Response 8: It was changed ‘’dry mass” to ‘’dry matter”. Line: 101

Comments 9: Line 80-82: these are physiological changes, not behavioural changes.

Response 9: It was move to earlier paragraph.

Comments 10: Line 82: lying, not sitting.

Response 10: It was changed ‘’sitting” to ‘’lying’’. Line: 103

Comments 11: Line 95: choice of words - At the beginning???

Response 11: It was changed „at the beginning” to „at the outset” line: 115

Comments 12: Line 119- 121: repeating what has been said earlier.

Response 12: Repeating was delete.

Comments 13: Line 138_140: Discussion of body temp would be better earlier, before discussion of changes in temp.

Response 13: In the introduction, general information about changes occurring during childbirth was provided, while in lines 138-140 already mentioned the use of specific devices to detect calving based on changes in temperature. Also added a discussion on body temperature.

Comments 14: Line 151-152: How does this sensor reduce the negative impacts of dystocia?

Response 14: It was explained how sensors reduce the negative impacts of dystocia. Lines: 182-185

Comments 15: Line 160- Measuring progesterone requires a blood samples - not possible on all farms.

Response 15: A daily blood draw from cows may not be possible on every farm, but as Avendano (2022) suggest you can do such a test every week.

Comments 16: Line 162-169: Is it a good idea to take strip samples before calving?

Response 16: Yes, this is an easy method to do in the field and by people less experienced. Lines: 202-206

Comments 17: Line 192: could this change in behaviour be automated.

Response 17: Artificial intelligence is used to detect the calving and posture of the cow. The use of cameras allows it to recognise the cow's whereabouts and analyse its movements. Lines: 231-236

Comments 18: Line 195 - 220: What are normal lying times in cattle?

Response 18: The normal lying time in cattle is 12-14 h. Line: 260

Comments 19: Line 262-263: changes in glucose concentration post calving - too late to be a predictor of calving?

Response 19: It was only mentioned about glucose changes before birth. Line: 304

Comments 20: Line 212: Borchers et al is reference 14.

Response 20: This has been corrected

Reviewer 3 Report

Comments and Suggestions for Authors

GENERAL COMMENTS:

This is a very interesting and generally well-written paper. In this study, the authors gathered information about both invasive and non-invasive methods for observing the signs of impending calving. The aim was to discuss both these invasive and non-invasive methods for monitoring calving signs. Correct identification and prediction of calving whether by visual behaviors only or using sensors are both challenging and important in beef and dairy farming, as it allows for effective monitoring and timely obstetric interventions. This study is well-suited for the Animal audience, providing a state-of-the-art review and direction for future research on the application of precision livestock farming in cattle peripartum period. However, as a review, it really lacks mention and discussion of some key studies on certain topics. For example, Cooper-Prado et al. (2011), Borchers et al. (2017), Ricci et al. (2018), Keceli et al. (2020), and Vicentini et al. (2021) offer valuable insights into the use of sensors for detecting and predicting calving in cows. Additionally, Lidfors et al. (1994), Kovács et al. (2017), Rice et al. (2017), Zehner et al. (2019), and Vicentini et al. (2022) discuss important behavioral aspects of the peripartum period, whether based on direct observations or by technological tools. Before being considered for publication, the paper needs a thorough literature review to ensure it comprehensively covers the state of the art on the main topic. I encourage authors to further explore the subject and expand their literature search to include other relevant studies. At the end I have listed some references that may be useful. A few comments and suggestions for the authors are outlined below

SPECIFIC COMMENTS:

Lines 54-60: Please merge these lines into a single paragraph.

Line 60: Please remove unnecessary “.” before references 14 and 16.

Line 60: I believe "fetal membranes" would be a more precise term than simply "placenta".

Line 65: Please remove unnecessary “(” or adjust accordingly.

Lines 67-68: Could you clarify whether you are referring to vaginal temperature or internal body temperature? Adjust accordingly.

Table 1: Please consider adding the indicated references (listed at the end) and specifying whether the studies were conducted on beef or dairy cows.

Lines 311-320: Repeated from Lines 302-311.

Conclusion: It would be beneficial for the authors to include a statement suggesting future research directions related to the main topic of the paper.

REFERENCES THAT MAY BE USEFUL:

Lidfors, L.M., Moran, D., Jung, J., Jensen, P., Castren, H. 1994. Behaviour at calving and choice of calving place in cattle kept in different environments. Applied Animal Behaviour Science. 42(1), 11–28. https://doi.org/10.1016/0168-1591(94)90003-5

Lammoglia, M.A., Bellows, R.A., Short, R.E., Bellows, S.E., Bighorn, E.G., Stevenson, J.S., Randel, R.D. 1997. Body temperature and endocrine interactions before and after calving in beef cows. Journal of Animal Science. 75, 2526–2534. https://doi.org/10.2527/1997.7592526x

Aoki, M., Kimura, K., Suzuki, O. 2005. Predicting time of parturition from changing vaginal temperature measured by datalogging apparatus in beef cows with twin fetuses. Animal Reproduction Science. 86, 1–12. https://doi.org/10.1016/j.anireprosci.2004.04.046

Cooper-Prado, M.J., Long, N.M., Wright, E.C., Goad, C.L., Wettemann, R.P. 2011. Relationship of ruminal temperature with parturition and estrus of beef cows. Journal of Animal Science. 89, 1020–1027. https://doi.org/10.2527/jas.2010-3434

Saint-Dizier, M., Chastant-Maillard, S. 2015. Methods and on-farm devices to predict calving time in cattle. Veterinary Journal. 205(3), 349–356. https://doi.org/10.1016/j.tvjl.2015.05.006

Kovács, L., Kézér, F.L., Ruff, F., Szenci, O. 2017. Rumination time and reticuloruminal temperature as possible predictors of dystocia in dairy cows. Journal of Dairy Science. 100(2), 1568–1579. https://doi.org/10.3168/jds.2016-11884

Rice, C.A., Eberhart, N.L., Krawczel, P.D. 2017. Prepartum lying behavior of Holstein dairy cows housed on pasture through parturition. Animals. 7(4), 32. https://doi.org/10.3390/ani7040032

Ricci, A., Racioppi, V., Iotti, B., Bertero, A., Reed, K.F., Pascottini, O.B., Vincenti, L. 2018. Assessment of the temperature cut-off point by a commercial intravaginal device to predict parturition in Piedmontese beef cows. Theriogenology. 113, 27–33. https://doi.org/10.1016/j.theriogenology.2018.02.009

Zehner, N., Niederhauser, J.J., Schick, M., Umstatter, C. 2019. Development and validation of a predictive model for calving time based on sensor measurements of ingestive behavior in dairy cows. Computers and Electronics in Agriculture. 161, 62–71. https://doi.org/10.1016/j.compag.2018.08.037

Keceli, A.S., Catal, C., Kaya, A., Tekinerdogan, B. 2020. Development of a recurrent neural networks-based calving prediction model using activity and behavioral data. Computers and Electronics in Agriculture. 170, 105285. https://doi.org/10.1016/j.compag.2020.105285

Vicentini, R.R., Bernardes, P.A., Ujita, A., Oliveira, A.P., Lima, M.L.P., El Faro, L., et al. 2021. Predictive potential of activity and reticulo-rumen temperature variation for calving in Gyr heifers (Bos taurus indicus). Journal of Thermal Biology. 95, 102793. https://doi.org/10.1016/j.jtherbio.2020.102793

Giaretta, E., Marliani, G., Postiglione, G., Magazzù, G., Pantò, F., Mari, G., Mordenti, A. 2021. Calving time identified by the automatic detection of tail movements and rumination time, and observation of cow behavioural changes. Animal. 15(1), 100071. https://doi.org/10.1016/j.animal.2020.100071

Vicentini, R.R., El Faro, L., Ujita, A., Lima, M.L.P., Oliveira, A.P., Sant’Anna, A.C. 2022. Is maternal defensiveness of Gyr cows (Bos taurus indicus) related to parity and cows’ behaviors during the peripartum period? PLoS One. 17(9), e0274392. https://doi.org/10.1371/journal.pone.0274392

Mota-Rojas, D., Bienboire-Frosini, C., Orihuela, A., Domínguez-Oliva, A., Villanueva García, D., Mora-Medina, P., Grandin, T. 2024. Mother–offspring bonding after calving in water buffalo and other ruminants: sensory pathways and neuroendocrine aspects. Animals. 14(18), 2696. https://doi.org/10.3390/ani14182696

Author Response

Dear Reviewers,

On behalf of my co-author and myself, I would like to thank the reviewers for their valuable comments and suggestions. Below are detailed responses to comments on the manuscript. The corrections made are highlighted in the manuscript.

The manuscript was also proofread for linguistic correctness. The language certificate is sent with the manuscript

Sincerealy,

Daria Wojewodzic, Marcin Gołębiewski, Grzegorz Grodkowski

Comments 1: Lines 54-60: Please merge these lines into a single paragraph.

Response 1: It has been applied.

Comments 2: Line 60: Please remove unnecessary “.” before references 14 and 16.

Response 1: It was removed.

Comments 3: Line 60: I believe "fetal membranes" would be a more precise term than simply "placenta".

Response 3: It was changed ‘’placenta” to ‘’fetal membranes’’

Comments 4: Line 65: Please remove unnecessary “(” or adjust accordingly.

Response 4: It was removed.

Comments 5: Lines 67-68: Could you clarify whether you are referring to vaginal temperature or internal body temperature? Adjust accordingly.

Response 5: It was referring vaginal temperature. Line: 88

Comments 6: Table 1: Please consider adding the indicated references (listed at the end) and specifying whether the studies were conducted on beef or dairy cows.

Response 6: They were added the indicated references to the Table 1: Aoki et al. (2005), Cooper-Prado et al. (2011) and to the Table 2: Rice et al. (2017), Giaretta et al. (2021) it has been specified which tools are used in dairy cows and which in beef cows in the Table 1.

Comments 7: Lines 311-320: Repeated from Lines 302-311.

Response 7: Repetition has been romoved.

Conclusion: It would be beneficial for the authors to include a statement suggesting future research directions related to the main topic of the paper.

It was included a statement of future research directions related to the main topic of the paper. Lines: 359-378 and 383-385. In addition, the text refers to new literature: Berglund et al. (2003), Nejash et al. (2016), Castro et al. (2011), Hue et al. (2021), Aoki et al. (2005), Cooper-Prado et al. (2011), Guliński et al. (2014), Grodkowski et al. (2023), Rice et al. (2017), Zehner et al. (2017), Hogeveen et al. (2010), Dominiak et al. (2017) and Keceli et al. (2020), Kovács (2017), Giaretta et al. (2021), Emam et al. (2023), Miwa et al. (2019).

Round 2

Reviewer 2 Report

Comments and Suggestions for Authors

Introduction: This section could do with some rationalisation. The new information presented in paragraph two means there is repetition between this paragraph and paragraphs three and 4.

Also the following should be corrected.

Line 38: Dystocia, a difficult, prolonged labor, that may require veterinary intervention. It can be caused by a narrow birth canal, weak contractions, the calf being too big

Line 54: to drink colostrum within 6 hours

Line 66: It is noteworthy that…..

Line 73: This should refer to endocrinological changes rather than physiological changes

Line 81: rumination time decreases, rather than decrease

In response to my previous comment 15: Line 160- Measuring progesterone requires a blood samples - not possible on all farms.

The authors provide this response 15: A daily blood draw from cows may not be possible on every farm, but as Avendano (2022) suggest you can do such a test every week

The Avendano (2022) reference is not cited in association with this discussion of changes in progesterone concentrations. Weekly progesterone concentrations would not be sufficient for this purpose.

Line 319: Cows kept at pasture, rather than housed at pasture

Line 321: shorter lying times rather than smaller lying times

Comments on the Quality of English Language

The English is generally OK but needs some revisions.

Author Response

Thank you for your comments.

Comments1: Introduction: This section could do with some rationalisation. The new information presented in paragraph two means there is repetition between this paragraph and paragraphs three and 4.

Response 1: The introduction has been re-edited, sentences changed, repetitions, grammar improved, as well as information that repeats.

Comments 2: Line 38: Dystocia, a difficult, prolonged labor, that may require veterinary intervention. It can be caused by a narrow birth canal, weak contractions, the calf being too big

Response 2: Amendments have been made to the text.

Comments 3: Line 54: to drink colostrum within 6 hours

Response 3: The amendment was added to the text.

Comments 4: Line 66: It is noteworthy that…..

Response : The amendment was added to the text

Comments 5: Line 73: This should refer to endocrinological changes rather than physiological changes

Response 5: It has been corrected to show that these are endocrinological, not physiological changes.

Comments 6: Line 81: rumination time decreases, rather than decrease

Response 6: The amendment was added to the text.

Comments 7:  In response to my previous comment 15: Line 160- Measuring progesterone requires a blood samples - not possible on all farms.

The authors provide this response 15: A daily blood draw from cows may not be possible on every farm, but as Avendano (2022) suggest you can do such a test every week

The Avendano (2022) reference is not cited in association with this discussion of changes in progesterone concentrations. Weekly progesterone concentrations would not be sufficient for this purpose.

Response 7: Using a rapid progesterone test is a quick and easy way to detect impending birth in dairy cows, especially in the field. This is indicated by Streyl et al. (2011). The use of these tests can help in the decision to start monitoring of birth in a cow, and is also used in veterinary and clinical management. Matsas (1993) indicated that progesterone tests are 90.0% effective at detecting impending birth within 12 hours. References are cited in association with this discussion. Lines: 192-206.

Comments 8: Line 319: Cows kept at pasture, rather than housed at pasture

Response 8: It has been corrected „housed” to „kept”.

Comments 9: Line 321: shorter lying times rather than smaller lying times

Response 9: It has been corrected „samller” to „shorter”

Reviewer 3 Report

Comments and Suggestions for Authors

I believe that after the revisions, the inclusion of important articles in the field (although not all of the suggested ones were included), and the improvements made to the discussion, the manuscript now fulfills the purpose of a review.

Author Response

comments: I believe that after the revisions, the inclusion of important articles in the field (although not all of the suggested ones were included), and the improvements made to the discussion, the manuscript now fulfills the purpose of a review.

Response:

Thank you again for your suggestions. We are pleased that the changes made have improved the quality of the manuscript.